# Pathophysiology, Diagnosis, and Management of Canine Intestinal Lymphangiectasia: A Comparative Review

**DOI:** 10.3390/ani12202791

**Published:** 2022-10-15

**Authors:** Sara A. Jablonski

**Affiliations:** Department of Small Animal Clinical Sciences, College of Veterinary Medicine, Michigan State University, East Lansing, MI 48824, USA; sjw@msu.edu

**Keywords:** canine, intestinal lymphangiectasia, protein-losing enteropathy

## Abstract

**Simple Summary:**

Canine intestinal lymphangiectasia (IL) is a disorder characterized by dilation, obstruction, and/or dysfunction of the lymphatic vessels within the small intestine. Dogs with IL often suffer from diarrhea, weight loss, vomiting, and fluid accumulations secondary to protein loss from the intestine. This review compiles the current knowledge of the pathophysiology, diagnosis and management of disorders of the intestinal lymphatic vasculature in dogs and humans and aims to (1) improve understanding of these disorders and (2) identify areas where research is needed to improve outcomes for dogs with IL.

**Abstract:**

Intestinal lymphangiectasia was first described in the dog over 50 years ago. Despite this, canine IL remains poorly understood and challenging to manage. Intestinal lymphangiectasia is characterized by variable intestinal lymphatic dilation, lymphatic obstruction, and/or lymphangitis, and is a common cause of protein-losing enteropathy in the dog. Breed predispositions are suggestive of a genetic cause, but IL can also occur as a secondary process. Similarly, both primary and secondary IL have been described in humans. Intestinal lymphangiectasia is definitively diagnosed via intestinal histopathology, but other diagnostic results can be suggestive of IL. Advanced imaging techniques are frequently utilized to aid in the diagnosis of IL in humans but have not been thoroughly investigated in the dog. Management strategies differ between humans and dogs. Dietary modification is the mainstay of therapy in humans with additional pharmacological therapies occasionally employed, and immunosuppressives are rarely used due to the lack of a recognized immune pathogenesis. In contrast, corticosteroid and immunosuppressive therapies are more commonly utilized in canine IL. This review aims toward a better understanding of canine IL with an emphasis on recent discoveries, comparative aspects, and necessary future investigations.

## 1. Overview of the Intestinal Lymphatic Vasculature 

The lymphatic system of the human and canine gastrointestinal tract plays crucial roles in the removal of interstitial fluid, transport of dietary fats, and immunoregulation. Despite these critical functions and contributions to intestinal homeostasis, historically the lymphatic vasculature has been poorly studied and a general understanding of lymphatic biology is lacking. In the last several decades, however, improved recognition of these critical functions has led to a surge in interest in the study of the lymphatic vasculature [1].Identification of specific markers for lymphatic endothelial cells (LECs) and novel imaging techniques have led to enhanced understanding of the functioning of the intestinal lymphatics in both health and disease, as well as the consideration of novel therapeutic targets for diseases involving the intestinal lymphatic vasculature [1,2].

Transport through the lymphatic system is unidirectional, designed to return fluid to the blood circulation. Interstitial fluid formed in tissues is first collected into blind-ended, non-contractile vessels known as lymphatic capillaries, or lacteals. In the intestine, lacteals are located exclusively in the villi, and typically reach 60–70% of the villus length, and in health, take up no more than 25% of the villus width. Lacteals are surrounded by a cage-like structure of venous and arterial blood capillaries. Collection of fluid into these capillaries is facilitated by LECs with discontinuous button-like junctions and overlapping flaps that function as unidirectional valves. Lymph fluid flows from the lacteals to a connected network of submucosal lymphatics at the base of the villi [2,3,4]. It is generally believed that lacteals and the submucosal lymphatic network do not contain smooth muscle, however, some studies suggest the presence of longitudinally oriented muscle cells associated with the lacteals [5], which may act to constrict the lacteal and propel fluid. The lymphatic network in the muscularis layer of the small intestine is seemingly not connected to the lacteals and submucosal lymphatic network, however, both systems drain into the lymphatic collecting vessels near the mesenteric border of the small intestine. The LECs of the lymphatic collecting vessels contain zipper-like junctions at the cell borders with no openings and are comprised of functional units known as lymphangions. Lymphangions contain smooth muscle and valves, allowing for spontaneous contraction and transport of lymph downstream to lymph nodes, the thoracic duct, and back into blood circulation [2,3] (Figure 1).

In addition to returning fluid to the blood vasculature, lymphatics are the primary transporter of lipid and lipid-soluble substances (including lipid-soluble vitamins) from the intestine to the blood. The specific molecular methods that regulate uptake of chylomicrons into lacteals are not well understood, however it is known that this task cannot be completed through portal blood absorption [1,6,7,8].

The lymphatic vasculature is critical to the transport of intestinal immune and inflammatory cells, and interactions between LECs and leukocytes have been demonstrated to influence immune cell migration, thus modulating the immune response. Thus, dysfunction in the lymphatics can lead to alterations of immune cell trafficking [3,7,9]. Lymphatic endothelial cells also appear to have roles in prenatal lymphatic patterning and post-natal control of formation of new lymphatics, or lymphangiogenesis [1,10,11,12]. Finally, the intestinal lymphatic vasculature is influenced by and an influencer of gut microbiota health, epithelial integrity and barrier function [9].

Lymphatic pumping is influenced by contraction of skeletal muscle, respiratory movements, pulsation of the blood vasculature, variations in central venous pressure, and a variety of intrinsic factors. Intrinsic factors that affect lymphatic pumping include cholecystokinin, glucagon, serotonin, dopamine, substance-P and bradykinin, all known to increase lymphatic pumping, and nitric oxide, vasoactive intestinal peptide, prostacyclin, acetylcholine, and anti-diuretic hormone, which decrease lymphatic pumping. Oxygen levels in lymphatics are relatively low, similar to venous blood, which may make lymphatics especially vulnerable to injury during ischemic stress [1,13].

## 2. Pathophysiology 

### 2.1. Lymphatic Disturbances and Consequences 

Disorders of human and canine intestinal lymphatics can include lymphangiectasia (pathologic dilation of lymph vessels), lymphatic vasculature obstruction, lymphangiogenesis (formation of lymphatic vessels from pre-existing ones), and lymphatic dysfunction [9]. Importantly, these changes can be present in all layers of the small intestine and the mesentery (Figure 1). Many human disorders of the intestinal lymphatics present early in childhood, and in dogs, many breed predispositions exist, both of which suggest disorders of the intestinal lymphatics can be primary or congenital. Interestingly, some intestinal lymphatic conditions that are considered congenital can present later in life than expected, and IL can sometimes be discovered incidentally, which may suggest that conditions can exist asymptomatically or until a stimulus exacerbates it. In one mouse model of lymphatic insufficiency, mice were clinically normal for 1.5 years until acute enteritis was chemically induced. Following recovery from acute enteritis, mice with lymphatic insufficiency had persistent inflammation as well as morphologic changes to their small intestine when compared with control mice [14]. This has led researchers and clinicians to suspect that lymphatic lesions may exist incidentally in some individuals until an inflammatory process exposes them.

Lymphangiogenesis can be induced by intestinal inflammation and is mediated by vascular endothelial growth factor (VEGF) [15]. It has been described in both experimental and clinical IBD, and largely is considered an adaptive rather than maladaptive change [9] though strategies to reduce lymphatic sprouting are sometimes successfully utilized in cases of IL. Effective lymphatic pumping and transport is altered in the presence of intestinal inflammation [16], but the mode of alteration and exact reasons why the lymphatics are affected is not well understood.

Additional mechanisms that are suggested to result in IL include increased hydrostatic pressure secondary to intestinal mucosal infiltrates or increased venous pressure at the level of the thoracic duct secondary to a variety of disorders [17,18,19]. Thus, it has long been understood that lymphatic abnormalities can occur as a consequence of intestinal inflammation, which, due to the functions of lymphatics, can worsen underlying disease processes.

Major consequences of lymphatic disturbances include hypoproteinemia, immunological deficiencies, and vitamin D deficiency, which can result in hypocalcemia and tetany. Hypoproteinemia often results in cavity effusions (peritoneal, pleural, and rarely, pericardial), and peripheral edema [9,12,20]. Immune abnormalities that have been associated with IL in humans include lymphopenia, hypogammaglobulinemia, and selective loss of CD4 cells [21,22]. The full implications of these quantitative immune abnormalities are not well understood, however, opportunistic infections have been reported in humans with primary intestinal lymphangiectasia (PIL) [23]. Immune suppression is sometimes pursued in cases of canine IL with PLE and could be considered counterintuitive or even harmful if a similar phenomenon occurs in dogs.

### 2.2. Intestinal Lymphatic Disorders in Humans 

Primary intestinal lymphangiectasia, eponymously referred to as Waldmann’s disease, results from malformed lymphatic vessels. It is commonly diagnosed in children prior to 3 years of age, however it can also present in adolescence or adulthood. In one review, the mean age of symptom onset was 13.3 years and 8.5 years until diagnosis [24]. However, one literature review found 49 cases from 46 case reports of PIL in adults in which onset of symptoms occurred after the patient’s 18^th^ birthday and median age at diagnosis was 43 [23]. Patients commonly develop hypoalbuminemia, hypogammaglobulinemia, lymphopenia, hypocalcemia, ascites, peripheral edema, gastrointestinal signs including diarrhea, and impaired development/growth. In 84 patients with PIL edema of the lower limbs (78%), diarrhea (62%), ascites (41%) were the most common clinical signs [24]. The most frequently observed biochemical abnormalities include hypoalbuminemia and hypogammaglobulinemia (73%), lymphopenia (63%), anemia (33%), and ionized hypocalcemia (25%) [23]. Histologically PIL is recognized as distended and obstructed lymphatics. Primary intestinal lymphangiectasia is chronic and debilitating requiring long-term therapy and can lead to life-threatening complications [20,25,26]. Intestinal lymphatic hypoplasia is a major differential diagnosis for PIL, though it is often apparent in infancy. This disorder is characterized by significantly fewer than normal lymphatics, with no obvious lymphangiectasia or lymphatic obstruction. It can be differentiated from PIL as patients with lymphatic hypoplasia typically lack lymphopenia, and the lack of lymphatics can be confirmed with the use of immunohistochemical labeling of lymphatic endothelial cells [20,27].

A variety of other syndromes of widespread dysfunction of lymphatics have been identified in humans, with varying effects on the intestinal lymphatic vasculature. Yellow nail syndrome (YNS) is a rare condition characterized by thickened yellow nails, lymphedema, and respiratory tract disease. The most accepted etiology is dysfunction of the lymphatic system, specifically abnormal lymphatic drainage. Reports of IL as a component of YNS have been described [28,29]. Noonan syndrome is a congenital condition of dysplasia of the lymphatics [30], and Nonne-Milroy syndrome is a hereditary form of lymphedema [31]. Klippel-Trenaunay syndrome is a rare congenital malformation of blood and lymph vessels [23]. Intestinal lymphangiectasia has been described in at least one case of Turner syndrome [32]. Additionally, IL has been reported in cases of Hennekam’s syndrome, which is a congenital autosomal recessive lymphedema, that also commonly features pleural lymphangiectasia and facial abnormalities [33].

An array of conditions can result in secondary IL in humans including connective tissue disorders, sarcoidosis, neoplasia, chromosomal abnormalities, intestinal volvulus, and inflammatory bowel diseases (IBD). It can also occur as a consequence of cardiac disease or portal hypertension [20,34]. Additionally, elevated venous pressure and subsequent dilation of lymphatics can occur in children following the Fontan procedure, a cardiac surgery performed to treat children with only a single effective ventricle [35]. Notably, lymphatic abnormalities have been increasingly recognized as contributing to the pathogenesis of IBD. Importantly, impaired lymphatic drainage due to dilation, obstruction, and dysfunction results in the accumulation of fluid and infiltrating immune cells, thus setting up chronic edema and inflammation [8,9,15]. In humans with IBD, increased lymphangiectasia, lymphadenopathy and lymphatic obstruction correlate with a poor prognosis. Whether the lymphatic abnormalities are a consequence or cause of the inflammation is debated, with some gastroenterologists even suggesting that lymphatic dysfunction is the root of Crohn’s disease (CD) [36]. Regardless, the importance of the intestinal lymphatic vasculature in fluid balance, the immune response, and even the health of the gut microbiota makes them worthy of further study in the pathogenesis and treatment of IBD.

### 2.3. Intestinal Lymphatic Disorders in Dogs 

Intestinal lymphatic disorders in dogs are less well characterized when compared to human disorders. Intestinal lymphangiectasia in dogs was first reported in 1968 [37] and is defined by varying degrees of lymphatic dilation with or without lymphatic obstruction and lymphangitis, and can be found diffusely, segmentally, or focally in the intestine. Hypoproteinemia develops due to loss of protein and lipid-laden lymph into the intestinal lumen, resulting in the syndrome of protein-losing enteropathy (PLE) [18,19,38,39,40,41,42,43]. A recent review of PLE syndrome in dogs reported that 214/469 (46%) dogs with PLE were diagnosed with IL [18]. Several breed predispositions to IL have been identified, including the soft-coated wheaten terrier, Chinese shar-pei, Norwegian lundehund, Maltese, Rottweiler, and Yorkshire terrier [17,18,44,45,46]. This suggests that IL may occur as a primary condition and/or a genetic susceptibility exists. However, IL in dogs can also occur as a consequence of another condition that results in altered lymphatic flow, such as chronic inflammatory enteropathies (CIE). Chronic inflammatory enteropathies are intestinal tract conditions characterized by clinical signs of GI tract disease of at least three weeks duration, the reasonable exclusion of infectious, neoplastic, mechanical, and extra-GI causes, and histopathologic documentation of intestinal inflammation. Importantly, CIE and IL occur together commonly in dogs with hypoproteinemia, with one study noting that 76% of dogs with CIE and low serum albumin had concurrent IL [47]. Additionally, dogs with lymphoplasmacytic enteritis (LPE), a form of CIE, and endoscopically visualized “white-spots” (thought to represent intestinal villi distended with lymph), were more likely to be hypoproteinemic than dogs without white spots, and the presence of white spots was correlated with lymphatic dilation histologically [48]. Other disorders that can result in IL through the alteration of lymph flow in dogs include neoplasia, such as intestinal lymphoma or adenocarcinoma, and although they have not been reported in the literature, theoretically as a consequence of pericarditis, pericardial effusion, right-sided heart failure, and portal hypertension.

Clinical signs of IL in dogs include diarrhea, weight loss, decreased appetite, and vomiting, though GI signs are absent in a small percentage of cases. In one study of 17 dogs, diarrhea, anorexia, lethargy, vomiting, and weight loss were the most common signs in 100%, 82%, 76%, 65%, and, 47%, respectively, [19]. Signs associated with hypoalbuminemia are common including ascites, respiratory difficulty or cough due to pleural effusion, and peripheral edema. Less common clinical signs include tetany or seizures secondary to hypocalcemia, and signs secondary to thromboembolic disease [17,18,38,49]. Two separate studies of a total of 39 dogs with IL report the most common biochemical abnormalities as hypoalbuminemia (87%), hypocalcemia (68%), hypoglobulinemia (54%), hypocholesterolemia (51%), and lymphopenia (46%). Hypocobalaminemia is also common [19,50]. Decreased serum concentrations of 25(OH)D are also common [51]. Histologic evaluation reveals lymphatic dilation, however other lesions are common including inflammatory infiltrates, dilated or cystic intestinal crypts, villous atrophy, and edema [50].

Focal intestinal lipogranulomatous lymphangitis (LGL) is a discrete form of lymphangiectasia reported in a small number of dogs. In one study, 8/10 reported cases occurred in French bulldogs [52]. These cases are distinguished from typical IL cases by the presence of a small intestinal mass often involving the adjoining mesentery. Masses are typically found in the distal jejunum or ileum, and histology reveals transmural granulomatous inflammation typically with extensive lipogranulomas that involve the mesentery and the muscularis and serosal layers of the small intestine [52,53,54].

Intestinal lymphatic hypoplasia has been suspected and reported in 3 dogs with PLE. Immunohistochemical labeling of lymphatic endothelial cells revealed a lack of villous lacteals in these dogs when compared to healthy controls [55]. Due to the small number of cases reported, the significance of this disorder is unclear, though it may be underrecognized. Finally, IL has been reported as a component of generalized lymphangiectasis in a Great Dane and associated with chylothorax in at least two dogs [56,57].

## 3. Diagnosis 

In both humans and dogs, definitive diagnosis of IL is made by histologic assessment of intestinal biopsies in a patient with compatible clinical signs and biochemical findings [17,18,20,34]. Direct visualization of the mucosa and various imaging modalities can also support the diagnosis. Advanced imaging of the intestinal lymphatics is much better described in humans vs. dogs.

The hallmark histologic finding in patients with IL is villous lacteal dilation, which can be described as mild, moderate or marked. In dogs, the World Small Animal Veterinary Association (WSAVA) issued guidelines that defines mild lacteal dilation as the central lacteal representing more than 25% but less than 50% of the width of the villous when sectioned longitudinally. Moderate lacteal dilation is when the central lacteal occupies 50−75% the width of the villous, and in marked lacteal dilation the lacteal dilates up to 100% of the villous. Marked villous lacteal dilation commonly causes a “club-shaped” appearance to the villous [58] (Figure 2). Patients with IL often have additional histologic lesions including various degrees and types of inflammatory infiltrates, crypt lesions, and other morphologic changes to the small intestine. Crypt lesions (dilated crypts filled with mucus and sloughed epithelium) were present in 5/17 (29%) of dogs with IL [19] and in 34/469 (7.2%) of dogs with PLE [18]. Concurrent inflammatory infiltrates are common as a consequence of lymph leakage and nonfunctional lymphatic vessels [47]. Because IL can also occur secondary to inflammation in the small intestine, determination of the primary or more significant process in patients with both IL and inflammation can be challenging. Intestinal lymphangiectasia may be limited to or worse in the ileum when compared to the duodenum in dogs, thus when IL is a differential it is important to obtain ileal biopsies where possible [59,60]. Importantly, IL can be diffuse, segmental, or focal, and though it usually affects the villous lacteal, in some cases it affects the deeper parts of the intestinal wall, such as the submucosa, muscularis, and serosal segments [50,60,61]. A study utilizing immunohistochemical (IHC) labeling of LECs in endoscopically obtained intestinal biopsies of dogs with PLE found that some dogs with lymphangiectasia in the proprial mucosa did not have accompanying villous lymphangiectasia (Figure 3). Thus, IL could be missed if only the villi are examined [60]. Similarly, in some cases of CD and PIL, dilated lymphatics are identified in the deeper mucosa, submucosa, and muscularis layers of the intestine, but not in the superficial mucosa/villous lacteals [62,63,64]. A more recent study evaluated for IL by labeling LECs with IHC in full-thickness small intestinal biopsies of dogs with lymphoplasmacytic enteritis (LPE) and lipogranulomatous lymphangitis (LGL). This study revealed IL in all layers of the small intestine, including the submucosa, muscularis, and mesentery, both in dogs with LPE and LGL. With the exception of one LGL case, dilated lymphatic vessels were observed in both the villus lacteal and deeper layers of the intestine, suggesting endoscopic biopsies should be sufficient to make the diagnosis in most cases [65]. However, the possibility still remains that IL can go unrecognized in superficial biopsies of the SI, and that it can be limited to areas of the intestine not routinely sampled endoscopically (e.g., jejunum). Despite this, obtaining biopsies endoscopically is still the preferred method in dogs with suspected IL, as it is considered safest, and is typically diagnostic. In humans with suspected PIL, use of IHC for diagnosis is considered routine, typically with D2-40+ antibody. Thus, consideration should be given to more widespread use of LEC labeling for accurate identification of the lymphatic vasculature in cases of suspected canine IL.

Diagnosing IL in endoscopically obtained intestinal biopsies has limitations. Thus, in some cases gross endoscopic exam, additional endoscopic techniques, and a variety of imaging techniques can be used to support the diagnosis. In dogs, endoscopic examination of the SI can reveal pinpoint to coalescing “white spots,” thought to represent dilated lacteals with or without lymphatic leakage (white streaks) (Figure 4). However, one study found that these lesions are only moderately sensitive and not specific for the diagnosis of IL [48] perhaps due to subjective interpretation. In humans, the intestinal mucosa in cases of PIL is commonly referred to as having a snow-flake appearance [64]. Interestingly, one study categorized PIL cases as “white-villi type” versus “non-white villi type” and compared their clinical characteristics and therapeutic responses. Investigators found that prior to treatment patients with ”non-white type” PIL had significantly worse serum albumin and fecal α-1-antitrypsin clearance compared to patients with endoscopically visible white-tipped villi. Patients with the “white-villi type” classification had lower serum immunoglobulin A and M concentrations compared to patients without endoscopically visible white-tipped villi. The corticosteroid response was better in the patients with the “non-white villi type” classification [66]. More recently, a group of investigators retrospectively classified the endoscopic features of 123 humans with PIL into four types: nodular-type, granular-type, vesicular-type, and edematous-type. Importantly, not all patients with a histologic diagnosis of IL had characteristic white-villus changes noted endoscopically. Histologically, patients with edematous type had lymphatic dilation in the submucosa, but no obvious lymphatic dilation in the lamina propria [67]. Similar classifications in the dog would explain why some dogs in the above mentioned IHC LEC study had IL in the submucosa but not in the lamina propria [60], and why some dogs without apparent histologic evidence of IL in endoscopic obtained biopsies appear to respond to therapy for IL [68]. However, further study is needed to evaluate whether similar discrete types may occur in canine IL.

Histologic diagnosis of IL in dogs with endoscopically obtained biopsies is limited to the section of SI that can be accessed and thus sampled. Double balloon enteroscopy (DBE) is a technique that allows extensive inspection of the small bowel. It is performed with the use of two balloons that are inflated and deflated in alternating sequence to allow the endoscope to progress through the SI. This technique is considered by some to be the gold-standard for diagnosis of PIL in humans with one study reporting diagnosis being obtained by gastroduodenoscopy in 60% of cases, ileocolonoscopy in 24% of cases and enteroscopy in 100% of cases [23]. In another study, endoscopy successfully diagnosed IL in 86% of cases, with the additional 14% of cases requiring video capsule endoscopy or enteroscopy for diagnosis [24]. Notably, enteroscopy can be accompanied by a higher rate of complications, including the possibility of post-procedural pancreatitis [69]. Double balloon enteroscopy has been reported in 14 laboratory dogs in two separate studies and was successful in the majority of cases with no complications encountered [70,71]. Further study is necessary to determine what patient population can be safely examined and biopsied with DBE and whether it has notable benefit compared to traditional endoscopy for the diagnosis of IL in the dog.

Video capsule endoscopy (VCE) has be used for complete evaluation of the small bowel in humans with suspected PIL [72,73,74]. In one case of a 14-month-old child with PIL, VCE discovered regional IL of the proximal jejunum to distal ileum that traditional gastroduodenoscopy and ileocolonoscopy did not reveal [73]. Other imaging modalities used to evaluate for IL in humans include scintigraphy, computed tomography lymphangiography (CTL), magnetic resonance lymphangiography (MRL), and dynamic contrast (DC) MRL [75,76,77,78]. Dynamic contrast MRL may hold the most promise. Importantly, injection of contrast into the distal limb lymphatic vasculature, inguinal lymph nodes or metatarsal paw pads in dogs is unlikely to adequately image the intestinal lymphatics as contrast will follow the path of least resistance and proceed anterograde through the progressively larger lymphatic vessels to the thoracic duct [76]. A study utilizing DCMRL in humans with PLE found that intra-nodal DCMRL was unable to demonstrate intestinal lymphatic abnormalities, however intrahepatic and intramesenteric DCMRL demonstrated enteric lymphatic abnormalities and leakage in the majority of patients. The authors concluded that DCMRL with multiple injection sites allows mapping of the abdominal lymphatic system [76]. In dogs with IL, transabdominal ultrasound may reveal hyperechoic mucosal striations, which are reported to be associated with lacteal dilation [79]. Oral administration of corn oil may help improve visualization of lacteals sonographically [80]. Technetium-labeled serum albumin scintigraphy has been used to localize protein loss in dogs with PLE, allowing for partial enterectomies to be performed in two dogs with focal disease [81]. The author attempted CTL via intra-metatarsal pad injection for visualization of the intestinal lymphatics in two dogs with histologically confirmed IL. The intestinal lymphatics, IL and or lymph leakage were not able to be visualized with this technique.

Elevated fecal α1-protease inhibitor levels or α1-protease inhibitor clearance can support the diagnosis in both humans and dogs. Serum and fecal canine α1- protease inhibitor concentrations have been demonstrated to reflect the severity of lacteal dilation in dogs. Serum-to-fecal α1-protease inhibitor ratio were lower in dogs with more severe lacteal and crypt lesions and more accurate in dogs with hypoalbuminemia [82]. Additionally, concurrent IL can be suspected in dogs with PLE and chronic inflammatory enteropathy (CIE) and lymphoplasmacytic enteritis (LPE) based on decreasing serum albumin, cholesterol and 25(OH)D concentrations [47,51,82,83]. Recently, serum C-reactive protein, bacterial lipopolysaccharide, and cleaved cytokeratin 18, and both serum and fecal zonulin were elevated in dogs with IL when compared to dogs with other gastrointestinal conditions [84]. Although nitric oxide is known to relax lymphatic vessels, there was no significant difference in inducible nitric oxide synthase between IL-positive and IL-negative tissues [85].

Finally, and crucially, the diagnosis of IL must include an investigation to determine whether the disease may be secondary to another cause. If another cause cannot be found, the assumption is the IL is the primary abnormality. Disorders that may result in secondary IL in the dog include local GI disorders such as chronic inflammatory enteropathies and neoplasia (e.g., intestinal lymphoma), and theoretically extra-GI disorders including portal hypertension, constrictive pericarditis, and right-sided heart disease.

## 4. Management 

If IL is suspected or confirmed to be secondary to another cause, initial management should directly address the underlying cause. Subsequently, monitoring should be performed to determine if the clinical syndrome associated with the IL has resolved with treatment of the purported cause. In some cases, however, it may be necessary to address the IL directly, even if it is considered secondary. In one prospective study evaluating response to a dietary adjustment in dogs with steroid-refractory inflammatory PLE, 8/10 dogs experienced a complete remission following dietary adjustment. Although the reason for the response is unknown, 7/8 dogs that responded were switched to a more fat-restricted diet than they had previously been consuming [68]. Thus, these dogs may have required therapy directed at IL in order to see clinical improvement from their primarily inflammatory PLE. In humans with Crohn’s disease, many studies describe concurrent lymphangiectasia, lymphangitis, lymphangiogenesis, and other lymphatic lesions [9,12,36,63]. Laboratory models suggest that normalization of lymphatic structure and function improves intestinal inflammation [8,14,16,86,87]. In one study, 61% of patients with CD not experiencing symptom relief despite various combinations of pharmaceuticals and dose escalations achieved clinical remission following institution of an exclusionary diet that was also low in fat and contained a moderate amount of soluble fiber [88]. Investigations are underway to determine if therapies directly targeting lymphatics may be useful in patients with Crohn’s disease.

### 4.1. Management of PIL in Humans

Lifelong adherence to a low-fat, high-protein, medium-chain-triglyceride (MCT) substituted diet is the cornerstone of therapy in humans with PIL. Dietary fat induces dilation of lymphatics in health. Therefore, decreasing dietary fat lessens the strain on diseased lymphatics and decreases excessive dilation reducing the risk of lymphatic rupture. Because they are absorbed directly into the portal circulation in humans, MCTs do not require the lymphatic system for absorption, making them ideal for the provision of adequate dietary fat and calories in a person with IL [23,89]. A literature search reviewing the evidence for MCT diet in patients with PIL found that 17/27 (63%) of MCT diet treated cases had complete resolution of symptoms compared to 10/28 (36%) of cases not treated with MCT. Feeding an MCT supplemented diet also improved mortality in this group of patients [90]. However, not all cases of PIL respond to dietary therapy alone. Information is limited on exactly what percentage of human patients with PIL are responsive to dietary changes alone, as most reports are of individual cases or small case series [91,92,93,94,95,96] (Table 1). Of thirty-eight cases of PIL treated with diet, 24 (63%) had clinical and biochemical improvement in response to diet, and children were more likely to show improvement when treated with diet when compared to adults [24]. In another study of 28 children with PIL, dietary therapy was successful in 79% of patients, with 6 children requiring additional therapies to achieve clinical control [97]. Patients responding to diet appear to largely require this therapy permanently, with one study reporting that clinical and biochemical abnormalities returned in patients following the withdrawal of low fat diet [98]. Partial or total parenteral nutrition (TPN) can be considered in patients failing to respond to traditional dietary changes, however one study showed enteral therapy to be an equally effective substitute for TPN in patients with IL [99]. Many gastroenterologists advocate for continuing this dietary approach in patients with PIL regardless of apparent response, as response may be inadequate but not entirely absent, and this disorder remains difficult to treat and associated with significant mortality. 

Second-line therapies in humans with PIL include radiologic interventions, surgery, and pharmaceuticals (Table 1). For focal cases of IL identified by scintigraphy or lymphangiography, surgical resection of affected bowel [100,101,102,103,104,105] and lymphatic embolizations have been utilized successfully. In one recent report, a 15-year-old male suffering from PIL for 7 years was treated with a series of two glue embolization’s of a leaking lymphatic channel in the duodenum. Following treatment, his serum albumin concentration improved from 1.9 g/dL to 5.0 g/dL over a period of 8 months [106]. Embolization is also commonly performed after the Fontan procedure or in patients with right heart failure and IL [64,107,108]. For more extensive disease, small numbers of reports have described positive responses to corticosteroids, octreotide, sirolimus or everolimus, propranolol [109,110], and tranexamic acid [97,111]. These therapies are often given in combination with dietary treatment. Corticosteroids have be used successfully in patients with IL as a complication of inflammatory disease, the Fontan procedure, but are not typically utilized in patients with PIL [112,113,114]. In patients with PIL and an incomplete response to diet, therapy with octreotide may be attempted. Octreotide is a long-acting somatostatin analogue that is thought to decrease intestinal fat absorption as well as inhibit gastrointestinal vasoactive peptides and stimulate the autonomic nervous system [23]. Multiple published case reports describe treatment of PIL with octreotide, sometimes as a sole therapy and other times in combination with other therapies [97,115,116,117,118,119,120,121,122,123]. While many of these case reports describe treatment as successful, octreotide may not be helpful in cases with extensive lymphangiectasia [64]. In those cases, sirolimus or everolimus may be attempted [124,125]. Sirolimus and everolimus inhibit mTOR, a key enzyme for cell growth and angiogenesis. Thus, sirolimus directly suppresses lymphatic sprouting and proliferation. Everolimus works similarly but has improved pharmacokinetic properties. Ozeki et al. [125] discovered that mTOR expression was increased in tissues affected by PIL and used everolimus to successfully treat a 12-year-old boy with PIL.

**Table 1 animals-12-02791-t001:** Summary of selected therapies for treatment of PIL in humans.

Therapy	Mechanism of Action	Citations
Low-fat, high-protein, MCT diet	Decreases excessive lymphatic dilation and rupture	[23,91,92,93,94,95,96]
Octreotide	Decreased fat absorption	[97,115,116,117,118,119,120,121,122,123]
	Inhibits GI vasoactive peptides Induces splanchnic vasoconstriction	
Sirolimus or everolimus	Suppress lymphatic sprouting and proliferation	[124,125]
Propanolol	Reduces expression of vascular endothelial growth factor	[109,110]
Tranexamic acid	Normalization of fibrinolytic activity (increased fibrinolytic activity leads to protein loss)	[97,111]
Surgical resection	Direct removal of affected tissue	[100,101,102,103,104,105]
Lymphatic embolization	Address focal leakage of lymphatic vasculature	[64,106,107,108]

Albumin infusions, supplementation of fat soluble vitamins, managing nutritional deficiencies are also commonly used in the management of IL in humans. Vitamin D is typically supplemented in the form of 25(OH)D, 1,5(OH)_2_D, alfacalcidiol, or vitamin D3.

### 4.2. Management of IL in Dogs

An important impediment to the proper management of IL in dogs is identification of the disease process and the recognition of its contribution to the clinical syndrome in the patient. In veterinary medicine, endoscopic exam and biopsy of the small intestine is commonly declined by the PLE dog’s guardian, and even when biopsy is performed IL can be missed or its contribution to the disease process underappreciated. Thus, it is critical for veterinary clinicians to know breed predispositions for IL and understand that significant negative correlations between lymphatic disease and serum albumin have been demonstrated in multiple studies even in cases where inflammation predominates histologically [47,60,83]. Therefore, IL should be considered a possibility in any dog suffering from PLE. Similar to Craven and Washabau [18], this author believes that because PLE is a life-threatening disorder with a high rate of mortality, the safest approach may be to assume all processes (ie, lymph fluid loss, increased intestinal permeability, mucosal injury) are occurring in a PLE patient and treat accordingly, in particular in severe cases or those not responding to therapy. It is also crucial that veterinary clinicians understand PLE as a heterogenous disease; a syndrome caused by a variety of disorders. Thus, individualized therapy is recommended rather than one standard approach.

When canine IL is confirmed or suspected, the most important part of the patient’s management is the implementation of a low-fat diet. There is no consensus on the definition of “low-fat” in veterinary medicine, however the fat content of current commercially available, highly digestible, low fat diets ranges from 17 to 26 g fat/Mcal ME (1.7 to 2.6 g/100 kcal). In dogs with moderate to marked IL, or dogs with refractory PLE, an ultra-low fat diet with < 15 g fat/Mcal ME (1.5 g/100 kcal) may be needed. In those cases, it will be necessary to provide a veterinary-nutritionist formulated home-prepared diet, which is formulated with consideration of the patient’s entire disease process and dietary fat content of the previous diet. Other important factors that a veterinary nutritionist would consider in cases of IL and PLE include fiber content, food volume, frequency of feeding and the food form [126]. Dietary choices are further complicated in cases of PLE where the patient is suffering from both inflammatory enteritis and IL and it is unclear which process is the driving force. These patients may benefit from a fat-restricted diet with a novel or hydrolyzed protein source, for which few commercial options currently exist. A board-certified veterinary nutritionist could help in the selection of a commercially available therapeutic diet, if one is thought to be suitable. In cases where a commercial diet is considered unsuitable, a home-prepared diet formulated by a board certified veterinary nutritionist may be the safest approach as the nutritionist can consider all the patient’s disease processes and their individual contributions to enteric protein loss. Several studies have reported a positive response to diet in canine patients with IL or heterogenous PLE that may include IL [49,68,127,128] (Table 2). Nagata et al. reported that dogs responding to diet as their only therapy for PLE generally had canine chronic enteropathy clinical activity index (CCECAI) of less than 8. Importantly, optimization of diet may be important even in dogs not initially thought to be diet-responsive, as one study reported improvement in clinical signs and serum albumin following a diet change in dogs with steroid-refractory inflammatory PLE [68]. Additionally, dogs should not be classified as unresponsive to food based on a single dietary trial, but rather multiple diet trials may be necessary. Similar to humans, dogs with IL might benefit from MCT since they are classically underconditioned but need to be fed a low-fat diet. However, the benefits of MCT in dogs with IL have not been critically assessed and they can affect palatability of the diet [126]. One study suggested that MCTs are still absorbed into the lymphatic system in dogs, therefore would not help to reduce lymph flow [129]. Assisted enteral feeding was significantly associated with a positive outcome in dogs with PLE [130]. The placement of an enteral feeding tube should be strongly considered in patients unwilling to accept recommended diets. 

Glucocorticoids are commonly prescribed to canine patients with both IL and PLE. It is important to note that glucocorticoids are very infrequently used in humans with IL, and there is no evidence for an autoimmune or immune-mediated etiopathogenesis in cases of IL. However, dogs with IL can develop lymphangitis and secondary enteritis that may benefit from anti-inflammatory doses of steroids, and some dogs with PLE may be suffering from both IL and immune-driven inflammatory enteritis [47]. In dogs with IL it is suggested to avoid immunosuppressive doses of steroids (>1 mg/kg, PO, q24), as some of the side effects associated with higher doses of steroids including muscle protein catabolism, hyperlipidemia, and thrombosis are particularly detrimental to dogs with PLE [18]. Though some clinicians may be tempted to increase doses of glucocorticoids due to concern for failure of drug absorption, a study performed in humans with Crohn’s disease found that adequate serum prednisolone levels were achieved despite concern for malabsorption. This study population included some patients who had a failure to improve despite treatment with prednisolone [131]. Since there is no suspected immune basis for IL, immunosuppressive therapies are not recommended or warranted, unless the dog is similarly afflicted by inflammatory enteritis suspected to be secondary to an overzealous immune system. It is pivotal to remember, however, that lymph is a local tissue irritant, so inflammation can occur as a consequence of IL and may not be the result of immune system dysfunction [19]. In one study of 43 dogs diagnosed with PLE, 14 of which had histologically confirmed IL, dogs treated with dietary therapy alone had significantly better outcomes than those treated with immunosuppressive therapy [132]. In another study of 31 dogs with PLE, 10 of which had lacteal dilation noted histologically, the addition of a secondary immunosuppressive agent to glucocorticoid therapy did not result in shorter time to improvement of albumin or improved outcome when compared to glucocorticoids alone [133]. 

Dogs with IL and secondary PLE should also be treated with supplementation of deficiencies (e.g., cobalamin), supportive therapies to improve appetite and combat nausea, if suspected, and thromboprophylaxis. Dogs with PLE are classified as at “high risk” for thrombosis based on the 2022 CURATIVE guidelines and thromboprophylaxis is recommended [134]. Until more information regarding the best thromboprophylactic approach in dogs with PLE is available, the use of clopidogrel or a factor Xa inhibitor (e.g., rivaroxaban) is reasonable. Similarly, the benefit of or best approach to treatment of hypovitaminosis D with or without ionized hypocalcemia, secondary hyperparathyroidism, and/or ionized hypomagnesemia in dogs with IL is unclear. Ongoing studies may help to clarify this. In the meantime, the use of calcitriol for dogs with IL and significant ionized hypocalcemia is recommended provided close monitoring of serum calcium, phosphorus, and 25(OH)D is performed. Canine albumin infusions, removal of problematic effusions, and therapies to address dysbiosis may also be considered on case-by-case basis. Colloids can be useful for oncotic support in patients that develop refractory effusions.

The use of other pharmaceutical therapies reported to be of benefit in human IL have not yet been published in dogs. However, octreotide has been used in dogs with IL by the authors and others at a 5−10 µg/kg, subcutaneously, q8−12 hours. If a positive benefit is noted, octreotide can be tapered to the lowest effective dose and frequency, and continued long-term. Anecdotal reports describe varying responses, with some dogs experiencing no improvement and others achieving apparent complete clinical and biochemical remission. More information is needed before this therapy can be routinely recommended in cases of IL. However, the use of octreotide can be considered in cases refractory or achieving incomplete responses to other therapies, including the previous administration of a novel protein, ultra-low fat diet.

## 5. Conclusions and Future Directions 

Despite being first described over 50 years ago, canine IL remains poorly understood with unacceptably high mortality rates. There is a need for research addressing various gaps in knowledge including ways to predict IL and improve diagnosis with the use of biochemical parameters and novel imaging and procedural techniques. Intestinal lymphangiectasia should be considered as a contributing factor in any case of canine PLE. Historically, immunosuppressive agents have been used in the management of IL despite limited evidence of their benefit in humans. Many recent studies support dietary management as the foundation of therapy. Investigations are needed to determine the best dietary approaches for individual cases of IL, and whether dietary treatment alone or dietary therapy in combination with anti-inflammatory glucocorticoids is superior in specific cases of IL. Investigations are also needed to explore the role of treatment of vitamin deficiencies (e.g., vitamin D) and to evaluate novel therapies (e.g., octreotide) in dogs with IL.

## Figures and Tables

**Figure 1 animals-12-02791-f001:**
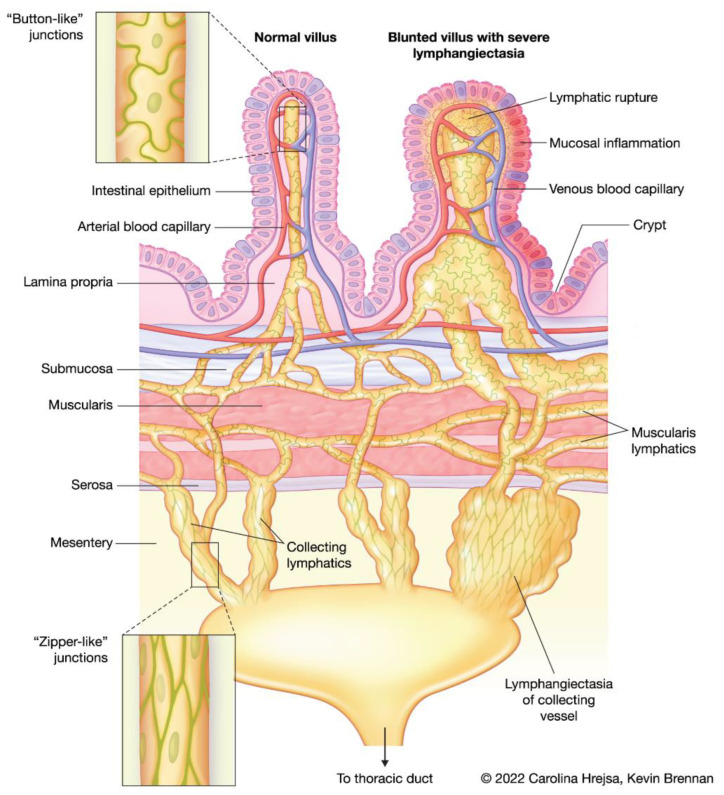
Schematic showing a normal villus with a healthy lacteal, cage-like structure of venous and arterial capillaries, and draining submucosal, muscularis, and collecting lymphatic vasculature. The differences between the cell junctions of the lacteal system versus the collecting lymphatics are depicted (insets). A blunted villus with severe lacteal dilation and lymphatic rupture, and lymphangiectasia of the draining lymphatic vasculature is shown for comparison.

**Figure 2 animals-12-02791-f002:**
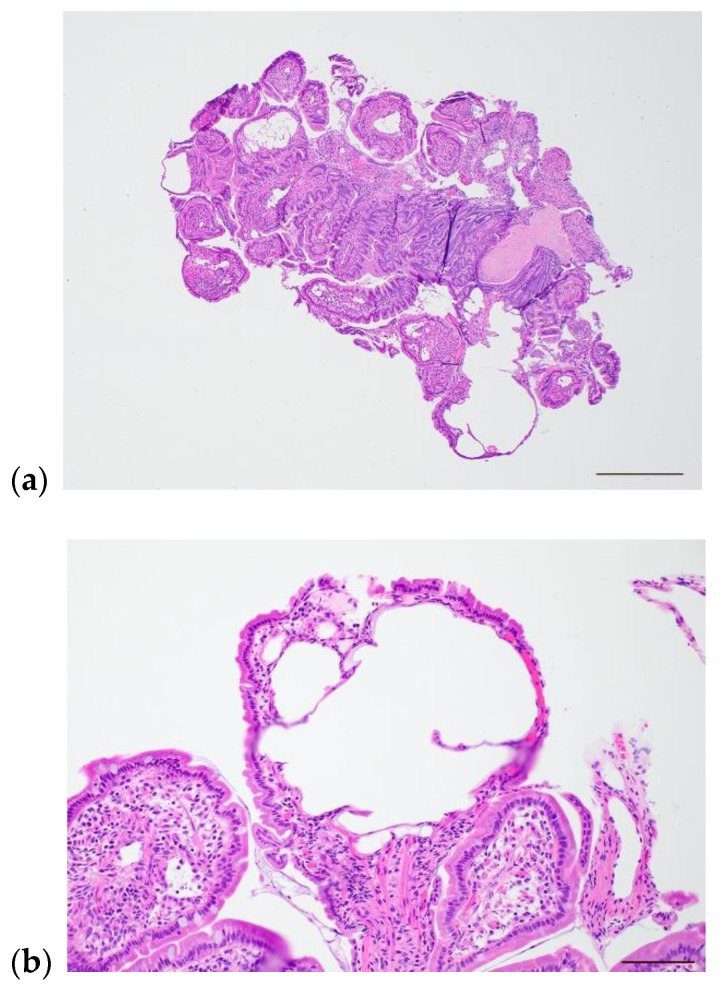
Photomicrographs of intestinal lymphangiectasia in a 5-year-old female, spayed soft-coated wheaten terrier. (**a**) Low-power image showing numerous moderate to markedly dilated lymphatics. Scale bar = 500 µm. (**b**) High-power image of markedly dilated villus lacteal. Scale bar = 100 µm.

**Figure 3 animals-12-02791-f003:**
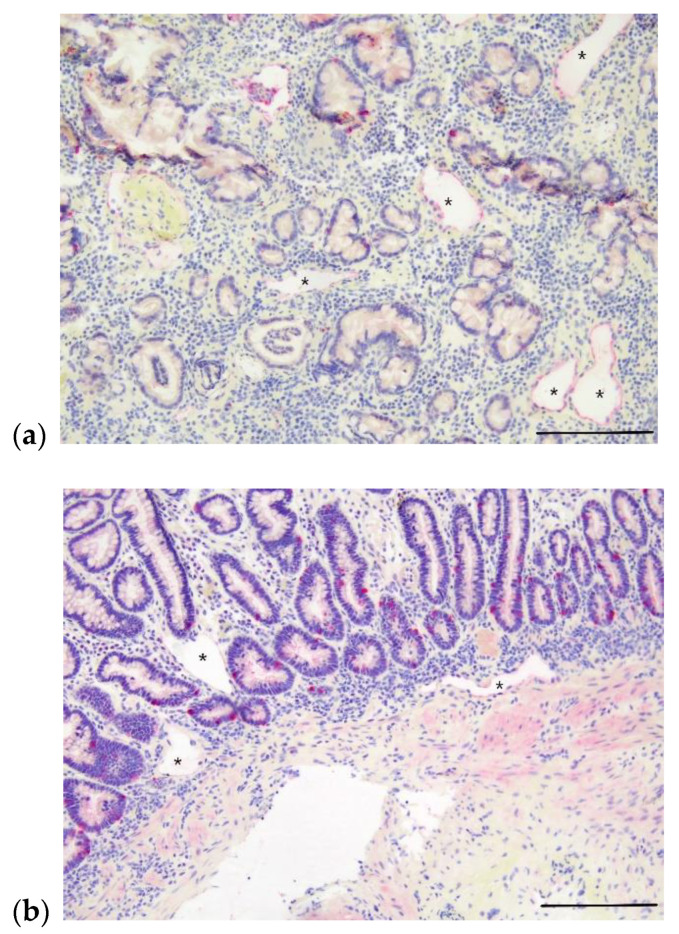
Photomicrographs of apparent lymphangiectasia in deeper layers of small intestine in two dogs with PLE without a diagnosis of lymphangiectasia on routine histologic examination. Intestinal lymphatics indicated by asterisks and immunohistochemically labeled with LYVE-1. (**a**) Apparent mucosal lymphangiectasia in a 9-year-old, male, castrated Australian shepherd dog with PLE; ectatic lacteals denoted by asterisks. Scale bar = 200 µm. (**b**) Apparent mucosal and submucosal lymphangiectasia in a 6-year-old, male, castrated Bernese mountain dog with PLE; ectatic lacteals denoted by asterisks. Scale bar = 200 µm.

**Figure 4 animals-12-02791-f004:**
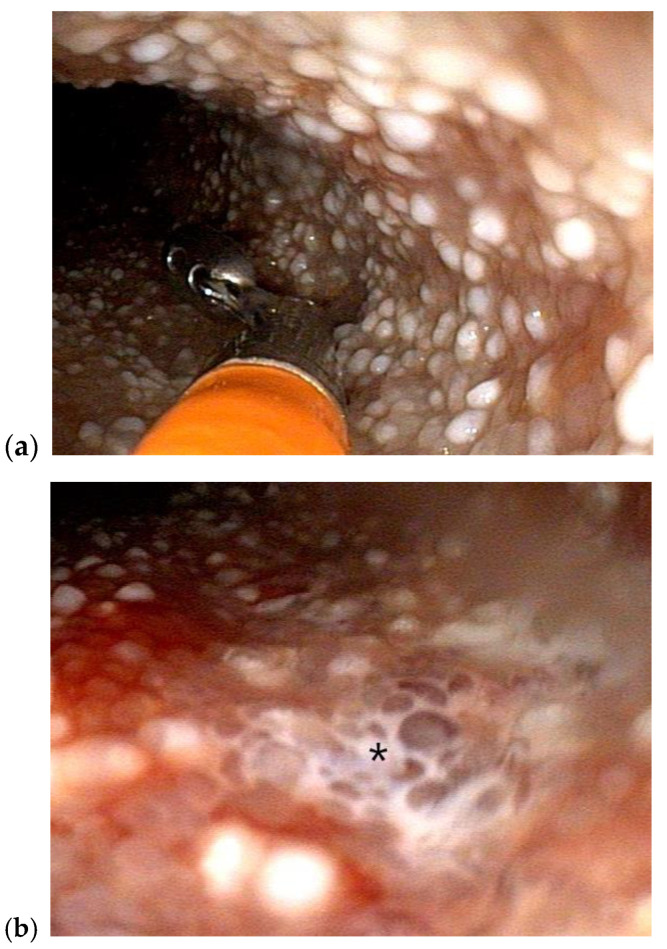
Duodenoscopy of a 5-year-old female, spayed soft-coated wheaten terrier with histologically diagnosed intestinal lymphangiectasia and the clinical syndrome of PLE (**a**) Multiple pinpoint to coalescing “white spots” consistent with dilated intestinal lymphatics are grossly visible during endoscopic exam and biopsy (**b**) leakage of lymph (shown with asterisk) is apparent on the mucosal surface of the small intestine.

**Table 2 animals-12-02791-t002:** Studies reporting on dietary treatment of suspected or confirmed IL in dogs.

Authors (Citation) Study Type	Dogs	Results
Okanishi et al. JVIM 2014;28:809–817 [49] Retrospective	24 dogs with unresponsive or relapsed histologically confirmed IL	19/24 (79%) dogs responded satisfactorily to dietary fat restriction
Rudinsky et al. JSAP 2017;58:103–108 [127] Retrospective	11 Yorkshire terriers with PLE, 4 with histologically confirmed IL	Clinical signs resolved completely in 8 dogs with dietary therapy alone
Nagata et al. JVIM 2020;34:659–668 [128] Retrospective	33 dogs with PLE, 25 with histologically confirmed IL	17/21 IL dogs treated with ultra-low fat diet had partial or complete response
Jablonski Wennogle et al. JSAP 2021;62:756–764 [68] Prospective	12 dogs with steroid-refractory PLE, 4 with histologically confirmed IL	8/10 dogs had complete remission with dietary change; 7/8 had dietary fat lowered
Olson and Zimmer. JAVMA 1978;173:271–274 [39] Case report	1.5 year old, female, Doberman pincher with IL and PLE	Remission of clinical signs and improvement of serum albumin with dietary therapy
Jones et al. NZ Vet J 1984;32:213–216 [40] Case report	8 month old, female, mixed breed dog with IL and PLE	Full recovery with dietary management alone

## Data Availability

Not applicable.

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
