# Peer review of "Pathophysiology, Diagnosis, and Management of Canine Intestinal Lymphangiectasia: A Comparative Review"

_animals, 2022, doi:10.3390/ani12202791_

Round 1

Reviewer 1 Report

Summary: The aim of the review is to describe the latest findings in the pathophysiology, clinical signs, diagnosis and management of intestinal lymphangiectasia in dogs; paralleling the disease present in people. Lymphangiectasia is characterised by a marked dilation and dysfunction of the intestinal lymphatic network, leading to the leaking of rich protein and lipid lymph into the intestinal lumen.  In dogs, it can be primary or secondary associated with chronic intestinal inflammation, neoplasia, or other extra-intestinal diseases such as cardiac disease and portal hypertension. The final diagnosis is made by a combination of clinical signs, clinical pathology, and histopathology. Treatment in people is mainly focused on diet management, whereas in dogs, diet and immunosuppressive treatment are often combined. Additionally, in contrast to people, the disease remains poorly understood in dogs and more studies are needed to unravel the mechanisms of this disease, its causes and treatment.

General and Specific comments

Thank you for conducting this review. The review is very complete, the parallel with humans is quite useful and provides an overall picture of a disease that is still poorly understood in dogs. There is a clear explanation of the structure and function of the intestinal lymphatic system, however, the review could benefit from some minor modifications. The simply summary mentions that the disease has a guarded prognosis with an approximately 50% survival rate. Is there data available about remission periods, medial survival time, and negative and positive prognostic factors?

Line 352: Could you include a brief description of the main findings, please?

Also, a recent study has been published about the association between intestinal lymphangiectasia and expression of inducible nitric oxide synthase in dogs with lymphoplasmacytic enteritis, although no association has been found (J Vet Med Sci. 2022 Jan; 84(1): 20–24).

Line 362: It should be figure 1, not 2.  In Figure 1, is figure b a higher magnification of a region from the figure a? If so, a bounding box in figure a denoting the region magnified in figure b, would be very useful.

Scale bars are not clear in figures 1 and 2 and the white balance has not been properly calibrated.

Spelling mistakes: Line 130 and 375: double period. Line 143: and instead of "wand". Line 138: location of the word “and” should be after biochemical findings, not before.

Reviewer 2 Report

This was a beautifully written comparative review paper, and a pleasure to read. Please see my (minor) suggestions below, and take or leave them!

Lines 44-46: Is this intestines of both humans and dogs? Might just want to clarify if in both or one species so reader knows from get go.

Lines 55-56: Should there be another comma after "however", so it reads "... network, however, both systems"...? I might be wrong, but just double check grammar.

Lines 83-131: Is this whole introduction section including pathophysiology from both species? I know it is broken down in the next few sections human versus dog, but might help to just clarify in this section if its just humans or a blend of what is known about pathophysiology of both.

Line 141: Think there should be an "and" prior to ascites here, so it should should read as "... diarrhea (62%), and ascites...".

Line 144: Would change the "it" on this line to "PIL" to be the most grammatically correct.

Line 263-264: Not sure that you need to say "IL" twice in this sentence. Think you could just say "Thus, IL could be missed if only the villi are examined."

Reviewer 3 Report

Dear Dr.  Jablonski Wennogle,

The paper is well written and does a good comparison between PLE/IL in dogs and humans.

This is a very passionate disease process for me and it was a pleasure reviewing this paper.

Some minor concerns/ suggestions

- Line 11: approximately 50% survival rate. I would like to see a refernce of this.

I do personally think that number is a bit high.

- Figure 1. - The picture would be clearer if it went a little step further and showed how the lymphatic rupture caused intestinal protein loss trough the  lymphatics going through the villi

- 192 - I do think there should be somemore expansion within the breeds here with respect to primary and secondary IL. 

Especially when in line 458 where there is a remark that it is 'critical for veterinary clinicians to know breed predispositions for IL'

I would have like to see Yorshire Terriers and Rottweiliers, SCWT primary PLE, versus the Shar Pei's and Norwegian Lundehunds, more like secondary etc. 

I do think from a clinical perspective, my approach to a YT and Rottweiler PLE is different from that of Shar Pei etc. This becomes more important from a therapeutic/ prognostic point clinically. 

I also think with LGL, breed disposition of the frech bull dogs etc are important. 

- 347, α1-antitrypsin inhibitor should be replacd by the newer term, especially when the latter reference (78) refers to it as  Α1-Proteinase Inhibitor

*- 357: To the best of my knowlegde PLE from heart disease in dogs has never been documeneted. If you have a reference that would be great. Disclaimer, this is a personal work of mine which was why the study was done

GI33

Fecal α1-Proteinase Inhibitor Concentrations in Dogs With Cardiac Disease

Joseph Cyrus Parambeth, Jordon P. Vitt, Jan S. Suchodolski, Jonathan A. Lidbury, Ashley B. Saunders, Joerg M. Steiner

https://onlinelibrary.wiley.com/doi/full/10.1111/jvim.13952

- 481: Just to stay neutral brands - I would add low fat kangaroo from Rayne, also consider a hydrolyzed low fat like Purina HA canned, RC HP canned, etc. 

*- 498 Assisted enteral feeding - I believe in this reference all but 1 were, E tubes. I am not sure using a G TUBE in a dog with low albumin is a good choice. There were some complications in the paper but n=1

(esophagostomy tube [19], gastrostomy tube [1], and nasogastric tube [1])

- I was surprised to read a PLE/ IL paper in dogs with no mention or reference to Intestinal Crypt Abscesses. Curious to know why this was not included in the paper at all. Ref 78 describes it.

On the clinical front, some internists treat non-responsive PLE with Intestinal Crypt Abscesses with antibiotics and they do see a benefit clinically, rather than just the dysbiosis/ tylosin tretament.

- 508 - While I understand that the recommendation, there is also a second school of thought higher doses may be needed as dogs with PLE/IL do not absorb the drug due to intestinal disease and sometimes parenteral administration may be necessary.

Otherwise the paper is awesome!

Sincerely 
